# Women’s Views of and Responses to Maternity Services Rendered during Labor and Childbirth in Maternity Units in a Semi-Rural District in South Africa

**DOI:** 10.3390/ijerph17145035

**Published:** 2020-07-13

**Authors:** Elizabeth Zitha, Mathilda M. Mokgatle

**Affiliations:** Department of Epidemiology and Biostatistics, School of Public Health, Sefako Makgatho Health Sciences University, Pretoria, Ga-Rankuwa 0208, South Africa; mmakgosizitha@gmail.com

**Keywords:** South Africa, labor and childbirth, verbal abuse, disrespectful care, midwife obstetric units, maternal care, interpersonal relations

## Abstract

Facility-based delivery is an important strategy to prevent poor labor outcomes, and midwives are at the center of maternal care. However, disrespectful and abusive maternal care by midwives is prevalent and leads to poor antenatal care utilization and increased numbers of home deliveries. The objective of the study was to assess the views of women about the care they received during labor and childbirth and describe the interactions between the women and the midwives. This was a qualitative study using in-depth interviews with women who had delivered in midwife obstetric units at a district hospital in Tshwane District, South Africa. Twenty-six women aged 18–41 years, and had delivered within the previous six months were selected, using purposive sampling. A thematic content analysis approach and NVivo11 computer software were used to identify emergent themes. Most women had had negative experiences of the maternity services they had received during labor and childbirth. Shouting and rude remarks by midwives caused tension between the midwives and the women and had created a major barrier for communication. The abuse and disrespect that the women were subjected to had created a hostile and uncaring environment for them. They felt stressed, fearful, and anxious throughout labor and childbirth. In response to the hostile environment, they employed manipulative tactics such as pushing before time in the hope of getting attention. These acts resulted in punitive responses from the midwives who joined forces against them, reprimanded, or ignored them. Good interactions, described as being respectful, approachable, and polite, and the sharing of information yielded positive experiences of maternity care. The state of maternity services rendered during labor and childbirth is counterproductive to the existing plan of increasing early antenatal care bookings and presentation to the facilities for labor and childbirth. There is a need to retrain midwives in the respectful care of women during labor and childbirth to facilitate a change in their attitudes.

## 1. Introduction

Midwives are the main providers of maternal child and women’s health in South Africa and sub-Saharan Africa, hence, a relationship of trust between the midwife and the woman in labor is vital [1,2]. Studies done on the quality of care and treatment during labor indicate how the quality of midwifery care can influence the outcome of labor [3,4]. According to the World Health Organization (WHO), globally many women still die due to complications related to pregnancy and childbirth. Sub-Saharan Africa accounts for 66% of these maternal deaths globally, with 596 deaths per 100,000 live births [5].

The maternal mortality rate (MMR) in South Africa had an overall decrease of 1152 from 2008–2010 to 339 fewer deaths in 2016 [6]. The institutional maternal mortality rate (IMMR) for possibly avoidable deaths declined to 83.3 per 100,000 live births. In 2014–2016 the MMR assessor’s report classified 61% of maternal deaths as avoidable, as they were mostly due to a poor quality of care during the antenatal, intrapartum, and postnatal periods [6]. About 39% of all maternal deaths with avoidable factors were related to a lack of skills in doctors, while 25% were related to a lack of knowledge in nurses. About a third of the avoidable maternal deaths were due to delays in referrals, failures to follow guidelines and procedures, and the mismanagement of obstetric abnormalities [6].

The literature also indicates that poor outcomes of labor are related to the abuse, disrespect, and neglect of the mothers, as well as a lack of monitoring in labor [7,8]. Studies conducted in sub-Saharan Africa and South Africa show that women go through many forms of disrespectful and abusive care (D&A), which includes physical and verbal abuse involving slapping, pinching, performing episiotomy and suturing without anesthesia, being shouted at, and being scolded [4,9,10,11,12]. A study conducted previously in Tshwane midwife obstetrical units (MOUs) showed that women experience care that is characterized by disrespect, neglect, and undesirable interactions with midwives during labor and childbirth [13].

The Guideline for Maternity care in South Africa indicates that healthcare workers have the responsibility to respect, care for, and be empathetic to women regardless of the poor working environment or supposed unsafe behavior of women during labor [14]. The literature indicates that women expect disrespectful and abusive care in healthcare facilities and regard this as the normal practice of midwives. Some women are convinced that the abuse and disrespectful behavior of midwives is a way of ensuring that they have safe deliveries and, therefore, never report the abuse [6]. Incidents of women being slapped or pinched for not pushing or not following instructions have been reported by some studies [6,8] The disrespectful and abusive maternal care in healthcare facilities continuously undermines the universal rights of child-bearing women, as outlined in the Maternal Care Charter [10,11,12]. Studies show that D&A result in women feeling humiliated, helpless, degraded, and worthless [6,15,16], while some studies have indicated the existence of a psychological impact of a negative birthing experience [8,15].

The literature shows that women in this district complain of negative experiences of care during labor. Due to this negativity, some women end up giving birth at home, and present at the facility only for the purpose of birth registration, while others do not attend antenatal care and present late in the facilities when labor is advanced [13].

Women’s negative experiences in the delivery room can have significance for later fear of childbirth. Therefore, it is important to critically evaluate the care during childbirth. There are global efforts to increase early uptake of antenatal care services to ensure safer facility-based childbirth, but there are significant barriers, such as perceived quality of care in some settings, preventing women from attending facilities [3]. While the few studies conducted in South Africa have focused on women’s experiences of D&A [3,13,16], the objective of this paper is to describe the interactions of women and midwives during childbirth and examine their responses to the care received. The findings of this study will help develop knowledge and understanding of the care women receive in labor and may be used for further research on these issues.

## 2. Methods

### 2.1. Study Design and Setting

A qualitative, explorative research design was used to interrogate about and describe the interactions of midwives and women during labor and childbirth in the MOUs of a semi-rural district in Tshwane, South Africa. Tshwane is a metropolitan municipality with a population of about 3.3 million, and is one of the five districts in Gauteng Province. Tshwane District consists of seven sub-districts which consists of 10 MOUs and five district hospitals. MOUs are part of the primary healthcare system. They offer low-risk antenatal and postnatal care services in addition to delivery services [14]. The study was conducted in one of the sub-districts, which consists of two MOUs and one district hospital to which the two MOUs refer patients with complications. The MOUs and the district hospital conducted 6179 deliveries in the 2018/2019 financial year. Ethical clearance was granted by Sefako Makgatho Health Sciences University Ethics and Research Committee (Ethical clearance number: SMUREC: SMUREC/H/57/2017:PG). All participants volunteered and gave a written consent to participate in the study. The setting for the study was one MOU and two primary healthcare clinics. The clinics receive postnatal women who delivered in the MOUs and in the district hospital. The study was conducted in partial fulfillment of the master’s degree of the lead author in July and August 2017.

### 2.2. Sampling and Recruitment

Purposive sampling was used to select women who had delivered within a six-month period in the care of midwives in the MOUs and the district hospital. Purposive sampling was performed to enable the researcher to use her own judgement to select participants who were knowledgeable about the topic [17]. In the current study only women who had delivered a live baby without complications and were above 18 years were included. Participants were selected when coming to the health facilities for postnatal care, the immunization of the baby, and family planning. Only women who had had normal vaginal delivery of live babies and were above 18 years old were included. Twenty-six women were interviewed, the sample being considered to be complete when new data no longer emerged from the data collection process and data saturation was reached.

### 2.3. Data Collection

The first author (E.Z.) conducted face-to-face in-depth interviews (IDIs) using a semi-structured interview schedule with open-ended questions. The interview schedule was self-developed in English and translated into Setswana, the local language. The main questions asked were: “What were the women’s experiences of care in labor?” and “What were their views about the care that they received in labor?” The researcher opted to use IDIs instead of focus group discussions because IDIs are more suitable for investigating sensitive topics like the experiences of labor and childbirth [18]. The in-depth interviews were conducted in a private room that was made available to the research for the interviews to ensure privacy. The researcher gave information to the potential participants regarding the purpose of the study and emphasized that participation was voluntary. Written informed consent was obtained from them before the data collection began. Confidentiality was respected by ensuring that the names of the participants and the names of the facilities were not recorded. The need to use an audio-recorder and to take notes was explained. Each interview lasted for about 30 min to 1 h. The interviews were performed in Setswana, except for one, which was performed in English. The interviews were recorded with the permission of the participants.

### 2.4. Data Analysis

Thematic analysis was done using the approach of Braun and Clarke (2016). First the audio recordings were transcribed verbatim and translated into English by the first author (E.Z.). The transcripts were read repeatedly and line-by-line coding was done with the second author (M.M.), who supervised the project, to identify the codes from five initial transcripts. A codebook was then developed in preparation for the software analysis. NVivo version 11 (QSR International, Melbourne, Australia), the qualitative data analysis software program, was used to apply coding to all the transcripts. The codebook was updated when new themes emerged. The emerging themes were discussed and refined by the authors and were used to present the data to be used in the presentation of the findings. See Figure 1 for illustration of the research process.

### 2.5. Ethical Approval

The researcher firstly obtained ethical clearance from the Sefako Makgatho Health Sciences University Research Ethics Committee (SMUREC/H/57/2017:PG). This was followed by asking for permission from the provincial department of the health and district research committee and the managers of the MOU at the two primary health clinics.

## 3. Results

A total of 26 women participated in the in-depth interviews. Most of them (23 out of 26) were between 20 and 30 years old. Most of them were para two (13 out of 26), had secondary education (24 out of 26), were single (19 out of 26), and were unemployed (17 out of 26).

### 3.1. Themes

Ten themes and six sub-themes emerged from the data analysis of the in-depth interviews. This paper will focus on only five themes that relate to the women’s views about the maternity services rendered during labor and delivery. These include (1) midwife/patient interactions, (2) verbal abuse and disrespectful care, (3) neglect and abandonment, (4) information sharing, and (5) responses to the care received.

### 3.2. Midwife/Patient Interactions

The participants had experienced their interactions with the nurses in different ways. Those who had a positive delivery experience were able to talk to the midwives without fear. A good interaction was promoted by good communication. In these circumstances the midwives were described as being friendly, approachable, attentive to their needs, provided information, and reassured them. This is indicated in the following statements:


*“She was helpful, and she was sweet. She was not pushy, she was not shouting, she was talking to me in a good way” (Otsile, 24 years, para1).*



*“… She told me that I was three centimeters dilated and she told me that I may go back to sleep because that blood was not a problem” (Lebo, 23 years, para 1).*



*“I was afraid to talk to her in the beginning, but I saw that she was good to me, so I started to be open for her as well … The communication was good, she was ever smiling,… encouraging me, saying you will finish, after the baby comes there will no longer have any pain” (Maggy, 22 years, para 2).*


The participants believed that in order to promote a good relationship they needed to follow instructions and not be arrogant towards the midwives. Some of the participants thought that the poor care was influenced by the way the woman behaved towards the midwives.


*“… I think if you cooperate and then they talk to you politely then I think you will also not become harsh, but if someone talks to you in a harsh manner, you will also start telling them off” (Roro, 29 years, para 3).*



*“I did my homework and found that it depends on how you are. I think if you are uncooperative you will get bad service, like if you are cheeky and you want to do things in your own way. I think they also get frustrated to a point that they would think that, this one wants to annoy us” (Rose, 27 years, para 2).*


### 3.3. Verbal Abuse and Disrespectful Care

The data reveal that the major hindrance to a good midwife–patient interaction was the verbal abuse and disrespectful care that the patients were subjected to. The midwives often said hurtful things to the women when they were shouting at them. Some participants felt that they were asked about and lectured about personal issues which had nothing to do with labor.


*“My complaint is that I was scolded like a small child just for asking them to put the baby on my chest, I just wanted that bond with my baby … From there while they were busy suturing me they were shouting at me because my legs were shaking” (Caro, 30 years, para 1).*



*“I think that some of the questions are not any of their business. Hei! Why do you give birth to so many children? Is it the same father? Is he going to take care of the baby? Is he working? I don’t like such thing” (Roro, 29 years old, para 3).*


As a result, some participants felt that they had had to be submissive. They pointed out that some midwives reported assertiveness or requests which were regarded as “knowing too much” to other midwives. The participants reported that the nurses told each other what had been said and then teamed up against them. This is reflected in the statements below:


*“… You should endure whatever they do to you. You must endure. Whatever they say, I must just keep quiet. So that at the end I should get help. Because if you answer them they will tell you, Yaa! On top of that you have a big mouth; we will leave you just there” (Johanna, 30 years, para 2).*


Some participants felt that they had received poor and disrespectful care because they were told to vomit in the medical waste bin which was smelly, while some were concerned about the lack of cleanliness and poor hygiene and felt that the midwives were exposing them to diseases.


*“They said pull that dustbin near you and vomit inside … it is not the one that you put normal waste in. So then I am inhaling the smell, the smell that comes from those things …” (Johanna, 30 years, para 2).*



*“Yoo! It is dirty, it is dirty, yoo! It is disgusting…. that place hai! Hai! Hai! There are toilets that are not working, it is smelly. It is not a conducive place to be used by pregnant women to bath, because even those who just gave birth they bath there, jaa…. And then the dishes that we use to bath are also not enough for everyone … There is no way that you can use the bath tub, it is dirty…” (Puleng, 20 years, para 1).*



*“…there are a lot of cockroaches, they were crawling on top of us, I don’t know why the cleaners are not cleaning …” (Julia, 24 years, para 2).*



*“… the bed that I was sleeping on the linen also had blood … No the blood was not mine, when I entered in the room I found blood … like felt bad and I asked myself that but why did they not at least change the sheet?” (Lebogang, 22 years, para 1).*


### 3.4. Neglect and Abandonment during Labor

Most of the participants who were dissatisfied complained about neglect and abandonment. Some were examined only when they arrived and were left in the room alone until they called for help. For some the call for help was ignored until the baby’s head was about to be delivered:


*“The baby was just going to come without anyone there. They were at the reception and I was in the room, which was closed with a curtain … I shouted, I must shout and said nurse! It is not good …” (Ncinah, 25 years, para 2).*



*“I called them from the chairs… Immediately after I climb on top of the bed I gave birth, and even then I had to also call them, they were not coming to examine me” (Kgantshi, 24 years, para 2).*



*“… but the other thing that I think was not right is that they leave us for a long time. They should come and check on us frequently …” (Roro, 29 years, para 3).*



*“Hai, Ok, but there was this other lady who gave birth on the floor… So she was coming back from the toilet, then she started screaming saying the baby’s head is coming out, Then the nurses went to her, and said hold it. They were supporting her so that she can go to her room, then the baby fell on the ground. That happened In front of the whole group of women” (Otsile, 24 years, para 1).*



*“So after that I pushed and delivered. I told her that … I am feeling dizzy. From there I did not see what happened. I just found myself on the floor. Then my front tooth broke” (Puleng, 20 years, para 1).*


Some participants complained that the nurses were asleep during the night and, hence, they neglected them. Some longed for the midwife’s presence in the labor room because they were afraid of being alone. This is indicated in the statement below:


*“… when I arrived she showed me the room and then she continued with her sleep. I came in at about twelve at night and then at about half past one my water broke. That is when she came to help me at half past one…. That is when she examined me, since I came in at twelve …. If I did not knock on her door, she would not have thought of waking up” (Lebo, 23 years, para 1).*



*“.. Not that she should babysit me, just at least so that you can feel that she is next to you… Why should they sleep for about three to four hours and remember it is at night … since they checked me at about 12 until at 5 in the morning” (Milly, 28 years, para2).*


The study revealed that some participants who delivered in the hospital were not attended to on arrival. Most of them were told to sit at the entrance of the labor ward, where there are chairs, and to wait to be attended to. Some pointed out that there was a queue that they had to follow to be attended to. Some said they understood that there was a shortage of beds, while others did not understand and were told not to wait. Others said their call for help was met with scolding and they were told to wait their turn, as they had just arrived. This is indicated in the quote below:


*“When I arrived they made me wait there for a long time.… I arrived at about 6 in the morning and at about 8 o’clock it was when they came to admit me and I had labor pain all along… They sat me at the benches there, there was nothing happening, they just left me there …” (Kgantshi, 24 years, para 2).*



*“Yes, you sit at the benches… when you come in and the beds are full, and you are having labor pain, you must wait for the other to finish first…” (Maggy, 22 years, para 2).*


Women who were admitted for the induction of labor were also made to sit in a passage near the labor ward, because there were no beds for them. The women said that they were uncomfortable and were made to care for one another.


*“We were so many on that day, like a lot, so we were sitting in the passage of the labor ward … That was the only place that had some space, because the labor ward was so full that day… but when the labor pain started to escalate, I was still at the chairs and it was getting late and I still did not have a bed at all…” (Otsile, 24 years, para 1).*


### 3.5. Information Sharing

The participants in this study stated that they got their information about labor from other women, their families, and previous experiences of labor. Some of the women were not given information about the progress of their labor or found out only after the examination had been done. The procedures were not explained.


*“No, you see those who helped me with my first labor, they were good. They were able to tell me that you are these centimeters dilated, and they would even show me the chart that, you see you get the baby when you are this far… That gave me the hope that at least the baby is on the way. Can you imagine when someone checks you and after that they don’t tell you how far you are, they just say you are fine. So you just don’t know what is happening in your process of labor” (Milly, 28 years, para 2).*



*“…Just to explain to you to show that this is your file. They don’t explain that as you are having high blood pressure this is what it means, no they are just busy writing … and you don’t even know what they are writing” (Noma, 27 years, para 2).*



*“They gave me an injection, and they also gave me certain two big pills, but they did not tell me what they were for… It makes me feel bad … I asked myself that, is my baby not fine or is there something wrong with me? It would have been better if they explained to me” (Lizzy, 37 years, para 3).*


### 3.6. Response to Care Received

The women responded to the care that they received in different ways. The following sub-themes emerged from this theme: Fear and uncertainty, panic and anxiety, and manipulation.

### 3.7. Fear and Uncertainty

The women who experienced their labor negatively also experienced feelings of fear and uncertainty. Some felt that delivering at home would have been better. This is reflected in the following quotes:


*“I could not understand. I was afraid. I wished to get out of that bed and run out of the facility and deliver at home…” (Caro, 30 years, para, 1).*



*“I was afraid, I don’t want to lie, because I felt like if I ask her questions, she will answer me in a rude manner because of she will be feeling sleepy… Maybe I would have lost my baby I don’t know...” (Tshego, 23 years, para 2).*


### 3.8. Panic and Anxiety

A 26-year-old participant who was having her second delivery explained about the psychological effects of the ill treatment that she got when she came for her first delivery. When she was supposed to deliver her second child she was afraid and stayed at home for as long as possible because she expected to find the same negative care in the facility. She thought that it would be better to develop complications at home, and then she would be admitted for that complication, because if it was just labor pains, they took you for granted and forgot you.


*“Here! You are spoiled; your mind is corrupted, yes… When I had my first baby, eei…! I was, I was, you know, I ended up seeing things, things which are not there [visual hallucinations] and I also had dreams, I would dream about things, scary things, I could see that I got this from here. I would panic and panic, just to panic without a reason, that when I saw that eeh…. This hospital! …” (Noma, 27 years, para 2).*



*“That is why most of the time we resort to staying at home so that labor may progress so that when we come in the labor ward they should find that I am towards the end. They will leave you in there, they won’t check you. After a long time they will say to you, you are going for an operation, meanwhile they left you there without being examined” (Noma, 27 years, para 2).*


### 3.9. Manipulation

Some participants in the study reported that when they felt that they were not being monitored they would employ tactics that would cause the midwife to attend to them. Some resorted to screaming, pushing, and asking to go to the toilet. This is reflected in the following quotes:


*“No! What I was thinking is that I should push, I think that’s when I will get help, because as long as you are sitting there, they are also relaxing… Yes, because except you calling them, saying can I please go to the toilet, can I please get this and that, they won’t come to check you” (Johanna, 30 years, para 2).*


Some women, who were told not to push before time because they were not yet fully dilated, pushed regardless of the midwife’s instruction not to push, which indicated a lack of trust in the midwife. This is reflected in the following quotes:


*“… I was busy pushing, and examined me and said, you should not push anymore your baby is still far. You see? I said Okay… I felt that the pain was causing me to push, so I pushed and pushed, I called her and she was not coming” (Puleng, 20 years, para 1).*


## 4. Discussions

The study explored women’s views of and responses to the maternity services rendered during labor and childbirth in maternity units in a semi-rural district in South Africa. The study found that women experienced both good and poor care during labor and childbirth in maternity units in the health district, even though negative experiences far outweighed the positive experiences. Good interaction between the woman and the midwife was important in facilitating a positive experience of labor. Good interpersonal relationships were characterized by respectful communication, the sharing of information about the labor and delivery, and the midwife’s being approachable and polite towards the woman. Similarly, previous studies conducted in the same district found that through good communication, the midwife could invite the woman to participate in her care without fear [9,13].

Being supportive and attentive to the woman’s needs during labor and childbirth made the woman feel cared for. When a woman felt cared for, she asked questions about the progress of the labor and the management plan, which enabled her to follow instructions and cooperate with the midwife. This created a relationship of trust and partnership, where both parties shared the responsibility of managing the process of labor. This is in keeping with previous findings [9].

As already said, some women experienced the interaction with the midwives negatively. The poor interaction was reflected in the verbal abuse, lack of patience, neglect, and lack of routine monitoring that the women were subjected to during labor and childbirth. Neglect and abandonment and a lack of routine monitoring leads to delayed care and may result in poor outcomes of labor [12]. It was found in the current study that women were often examined on admission and then left unattended until the very end of the process. Some gave birth unattended, when their calls for help were unheeded. Similar findings were reported in other studies [3,4,8,13,18]. One woman gave birth standing up, and the baby fell on the floor, due to a lack of monitoring.

The lack of support and the negative experience of a woman during labor and childbirth damages the trust between the woman and the midwife and impacts on the decision regarding future delivery in a health facility [19,20,21,22]. In the current study, as in other studies, the participants indicated that they would opt for giving birth at home in the future, where they felt that they would be supported [3,4]. The study found that some of the neglect and abandonment that the women experienced occurred at night. The women’s narratives indicated that the midwives sleep during the night and if they have to wake up to attend to the needs of the women, they would be annoyed, impatient and angry, and shout at the women. These finding are in line with those of previous studies in South Africa and other similar settings [3,4,20].

Verbal abuse in the form of shouting and rude remarks was the most prevalent form of D&A. Oosthuizen et al. (2017) also found that midwives constantly shout at the women during labor and childbirth. This verbal abuse caused tension and was a major barrier to communication between the women and the midwives. The current study, like others, found that midwives often say hurtful and judgmental words to their patients to embarrass, ridicule, belittle, and shame them during childbirth. Some women were shouted at for being too young to have a child while others were shouted at about having too many children from different fathers [1,8,15].

It was found in this study, as in other studies, that women who experienced disrespectful care and verbal abuse were stressed, anxious, and fearful throughout their experience of labor [3,15]. The negative and abusive interactions created an uncaring and hostile environment, which was characterized by the women’s lack of trust in the midwives during labor and childbirth. A 30-year-old woman delivering her first child indicated that, should she get pregnant again by mistake, she would have an abortion rather than experience another painful and hostile delivery.

In an uncaring and hostile environment, the women were expected to endure the treatment they were subjected to by the midwives. The study found that if they asserted themselves they were victimized and the midwives would “team up against them”. Orpin et al. (2018) and Bohren et al. (2015) reported that women’s assertive behavior often leads to conflict between the midwife and the woman. In this study, as in previous studies, midwives mistreated women because they were less informed and empowered to do anything given their low level of education, being in a semi-rural area, and being of a low socio-economic status. Furthermore, some of these women had no insight into how to report D&A, if they wished to, while others felt that complaints of D&A were not attended to [8,13,23].

Although the women had been taught about the signs of labor during antenatal care (ANC), they were not informed about the process of labor. Yet, they were expected to know what was expected of them and to follow the rules in the labor ward. The few who had information had acquired it from previous experiences of labor and childbirth or from other women. The study found that the information that women shared among themselves was often inaccurate and unsafe. For example, some of the women believed that the hours of labor were determined by how good the midwife was. They responded to the lack of information, the verbal abuse, the hostility, and the neglect from midwives by having a lack of trust, being fearful, and by being uncooperative. Other studies similarly found that women who lacked knowledge were fearful, reluctant to follow instructions, or even refused to permit vaginal examinations [1,24].

As reported in previous studies, women feel uncertain, unsafe, and fearful under the care of midwives who do not explain procedures or do not ask for consent before examining them during labor [25]. The lack of explanations leads to their feelings of detachment from the whole experience, as they simply watch as the midwives take control over everything [25,26]. One woman gave an account of how fearful she was of the spitefulness and maltreatment that the midwives were capable of giving. This led to the woman experiencing symptoms of postpartum psychosis. Orpin et al. [16] describe some of the negative experiences of care that women experience as psychological abuse.

### Limitations

There may have been possible bias on the part of the persons who did the analysis and the interpretation of the data. There were limited data on the characteristics of the heathcare facilities and the care providers. However, the strength of the study is that the quality of maternity care was described from the viewpoint of women who experienced it firsthand.

This study was based on face-to-face interviews with 26 women accessing maternity services from the primary healthcare clinics. Hence, the findings cannot generalize for all the clinics and the private sector maternity services. In a qualitative approach, the open-ended nature of the questions in the interview guide can be seen as a limitation. While it encourages discussion of themes of interest for the women being interviewed, some experiences of general interest might have been missed.

## 5. Conclusions

Most of the women had negative experiences of the maternity services they received during labor and childbirth. Shouting and rude remarks by midwives caused tension between the midwives and the women and created a major barrier to communication. The abuse and disrespect that the women were subjected to during their labor and childbirth created a hostile and uncaring environment for them. The women felt stressed, fearful, and anxious throughout their labor experience. Of significance is that the hostile environment, neglect, and abandonment led to the women to plan to avoid using state maternity services in future pregnancies. In response to the hostile environment, the women employed manipulative tactics such as pushing the baby before time in the hope of getting attention. The manipulative acts of the women resulted in punitive acts from the midwives, who joined forces against the women, reprimanded them, and ignored them.

The attitudes and behaviors of midwives are the major barrier to communication and the primary driver of disrespect and abuse during labor and childbirth. Therefore, it is imperative that the health districts develop measures to assess and enforce compliance with standards of quality care in maternity, consistent with the Guidelines for Maternity Care in South Africa and other international guidelines such as the better birth initiative strategy of mother- and baby-friendly birthing facilities. This should include the retraining of midwives in the respectful care of women during labor and childbirth, and to facilitate a change in the attitudes. 

There is a need to address the poor communication that is prevalent during labor and childbirth. The antenatal classes should include sessions that educate women about the labor process, the procedures that are done during labor, pain management, and what to expect from the midwives. Greater effort should be made towards changing the women’s negative attitudes towards facility-based childbirth.

## Figures and Tables

**Figure 1 ijerph-17-05035-f001:**
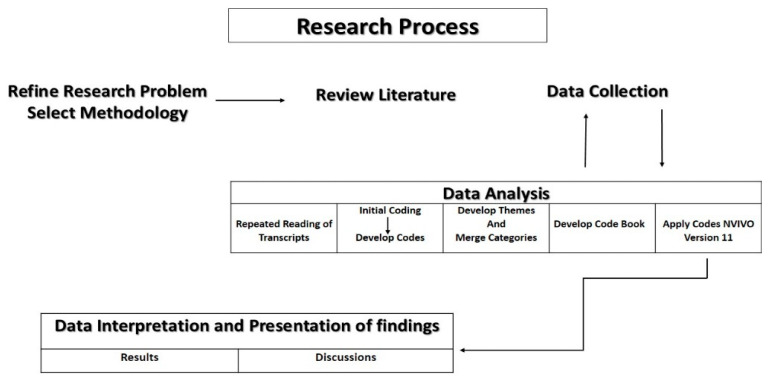
Research process.

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
