# Peer review of "Women’s Views of and Responses to Maternity Services Rendered during Labor and Childbirth in Maternity Units in a Semi-Rural District in South Africa"

_ijerph, 2020, doi:10.3390/ijerph17145035_

Round 1
Reviewer 1 Report
This is an interesting qualitative research that interrogated and described the interactions of midwives and women during labour and childbirth in the midwife obstetrical units of a semi-rural district in Tshwane, South Africa. Authors concluded that retraining midwives in the respectful care of women during labour and childbirthis was required and they need to facilitate a change in their attitudes. I have a few major and other minor concerns.
--The IJERPH abstract should be a single paragraph and should follow the style of structured abstracts, but without headings. Authors should revise the abstract.
--‘Plain English summary’ was not required. Please deleted it or moved to other manuscript section (e.g. Introduction).
--Authors should mention the specificity and importance of the research. It was still unclear in the Background section.
--I suggest authors to provide the diagram for relevant study procedures. Flow chart could help readers to realize the research.
--Qualitative research has some strength and limitations. I suggest authors to address relevant information in the Discussion.
-- Conclusions should be revised. It was hard to read because author split this section into many paragraphs.
Author Response
Reviewer 1
Comments and Suggestions for Authors
This is an interesting qualitative research that interrogated and described the interactions of midwives and women during labour and childbirth in the midwife obstetrical units of a semi-rural district in Tshwane, South Africa. Authors concluded that retraining midwives in the respectful care of women during labour and childbirth is was required and they need to facilitate a change in their attitudes. I have a few major and other minor concerns.
Point: --The IJERPH abstract should be a single paragraph and should follow the style of structured abstracts, but without headings. Authors should revise the abstract.
Response: The abstract is revised in one paragraph without subheadings.
Point: --‘Plain English summary’ was not required. Please deleted it or moved to other manuscript section (e.g. Introduction).
Response: Plain summary is removed
--Authors should mention the specificity and importance of the research. It was still unclear in the Background section.
Response: Specificity and importance included in the background (line 76-80)
--I suggest authors to provide the diagram for relevant study procedures. Flow chart could help readers to realize the research.
Response: A figure of the research process is included (line 85-87)
--Qualitative research has some strength and limitations. I suggest authors to address relevant information in the Discussion.
Response: I address the strengths and limitations under the section 4.1 in the discussion (line 381-386)
-- Conclusions should be revised. It was hard to read because author split this section into many paragraphs.
Response: I reorganized and rewrote the conclusion into paragraphs.
Reviewer 2 Report
Thank you for the opportunity to review this article. I am a qualitative researcher and always appreciate seeing the work of other qualitative researchers. There needs to be more of us in the world of academia. The conclusions of your article were very disturbing and I hope that some reform can be done. My suggestions for improvement are listed below.
Some grammatical errors need to be corrected. For example:
Line 18 Twenty-six women aged 18-41 years and had who delivered within the previous six months were (needs rewording)
Line 22 rude remarks by had midwives caused tension (use of had)
Line 267 So you just don‟t know what (use of apostrophe)
Line 292 and then you wouldl be admitted (spelling)
Line 341 giving birth at home in future, (in the future)
Line 380 not ask for consent before being examining them (before beginning? examining them)
Punctuation errors:
Lack of punctuation throughout the literature review and discussion section.
Lack of paragraph formation in conclusion. Either rewrite in paragraph form or show a list or bullets.
Other suggestions:
Line 317 and 318 “The study found that women experienced both good and poor care during labour and childbirth in maternity units in the district.” This opening statement of the discussion section makes the reader believe that there was equal amounts of positive and negative interactions reported by the participants. However, examples of these positive experiences were very few when discussing the themes. Either support these assertions of positive interactions with more quotes from participants or acknowledge that the negative experiences far outweighed the positive as the theme analysis seems to support.
Author Response
Reviewer 2
Comments and Suggestions for Authors
Thank you for the opportunity to review this article. I am a qualitative researcher and always appreciate seeing the work of other qualitative researchers. There needs to be more of us in the world of academia. The conclusions of your article were very disturbing and I hope that some reform can be done. My suggestions for improvement are listed below.
Some grammatical errors need to be corrected. For example:
Point: Line 18 Twenty-six women aged 18-41 years and had who delivered within the previous six months were (needs rewording)
Response: Placed a comma after years, and removed “who”
Point: Line 22 rude remarks by had midwives caused tension (use of had)
Response: Removed had
Point: Line 267 So you just don‟t know what (use of apostrophe)
Response: Removed the apostrophe
Point: Line 292 and then you wouldl be admitted (spelling)
Response: wouldl replaced with would
Point: Line 341 giving birth at home in future, (in the future)
Response: Include the in “the future”
Point: Line 380 not ask for consent before being examining them (before beginning? examining them)
Response: Removed being from the sentence.
Punctuation errors:
Point: Lack of punctuation throughout the literature review and discussion section.
Response: punctuation and grammatical corrected in the literature review, discussion and the introduction section.
Point: Lack of paragraph formation in conclusion. Either rewrite in paragraph form or show a list or bullets.
Response: I re-arranged and rewrote the conclusion in paragraph form and improved the grammar and punctuation.
Other suggestions:
Point: Line 317 and 318 “The study found that women experienced both good and poor care during labour and childbirth in maternity units in the district.” This opening statement of the discussion section makes the reader believe that there was equal amounts of positive and negative interactions reported by the participants. However, examples of these positive experiences were very few when discussing the themes. Either support these assertions of positive interactions with more quotes from participants or acknowledge that the negative experiences far outweighed the positive as the theme analysis seems to support.
Response: I mentioned that the negative expereiences outweighed the positive experiences. (line 308)
Round 2
Reviewer 1 Report
Thank you for the authors tried to answer the questions and revised the manuscript appropriately. I have reviewed the author’s responses one by one including other reviewers. The authors have answered all the questions that were raised by me. Therefore, my recommendation now would be to accept.